# A Small De Novo CNV Deletion of the Paternal Copy of *FOXF1*, Leaving lncRNA *FENDRR* Intact, Provides Insight into Their Bidirectional Promoter Region

**DOI:** 10.3390/ncrna9050061

**Published:** 2023-10-09

**Authors:** Przemyslaw Szafranski, Paweł Stankiewicz

**Affiliations:** Department of Molecular & Human Genetics, Baylor College of Medicine, Houston, TX 77030, USA

**Keywords:** gene regulation, divergently transcribed mRNA-lncRNA gene pairs, ACDMPV, developmental lung disorders, hypoplastic left-heart syndrome

## Abstract

Pathogenic single-nucleotide variants (SNVs) and copy-number variant (CNV) deletions involving the *FOXF1* transcription factor gene or CNV deletions of its distant lung-specific enhancer are responsible for alveolar capillary dysplasia with misalignment of pulmonary veins (ACDMPV), a rarely diagnosed lethal lung developmental disorder in neonates. In contrast to SNVs within *FOXF1* and CNV deletions involving only the *FOXF1* enhancer, larger-sized deletions involving *FOXF1* and the adjacent, oppositely oriented lncRNA gene *FENDRR* have additionally been associated with hypoplastic left heart syndrome and single umbilical artery (SUA). Here, in an ACDMPV infant without any congenital heart defect or SUA, we identified a small 5 kb CNV deletion that removed the paternal allele of *FOXF1* and its promoter, leaving *FENDRR* and its promoter intact. Reporter assay in the IMR-90 fetal cell line implied that the deletion may indeed not have significantly affected *FENDRR* expression. It also showed a polarization of the *FOXF1*-*FENDRR* inter-promoter region consisting of its ability to increase the transcription of *FENDRR* but not *FOXF1*. Interestingly, this transcription-stimulating activity was suppressed in the presence of the *FOXF1* promoter. Our data shed more light on the interactions between neighboring promoters of *FOXF1*-*FENDRR* and possibly other divergently transcribed mRNA-lncRNA gene pairs.

## 1. Introduction

Heterozygous loss-of-function of the mesenchymal forkhead transcription factor gene *FOXF1* at chr16q24.1 has been found in 80–90% of neonates with a lethal lung developmental disorder, alveolar capillary dysplasia with misalignment of pulmonary veins (ACDMPV, MIM 265380) [1,2]. Analyses of 34 ACDMPV-causative overlapping copy-number variant (CNV) deletions, leaving *FOXF1* intact, enabled us to define the distant, ~60 kb, lung-specific *FOXF1* enhancer region mapping ~286 kb upstream to *FOXF1* [3]. Recently, we proposed bimodal structure and functional parental dimorphism of this enhancer, with its Unit 1 acting stronger on the paternal chr16 and Unit 2 being more active on the maternal chr16 [4]. We also reported that transcription of not only *FOXF1* but also the adjacent, oppositely oriented lncRNA gene *FENDRR* is regulated by this enhancer [5]. The region between *FOXF1* and *FENDRR* is ~1.7 kb in size, GC-rich, and contains the promoters of both genes, making it transcriptionally bidirectional.

Eukaryotic promoters are intrinsically bidirectional featuring two transcription start sites; however, transcription in one direction usually prevails [6]. Nevertheless, as much as 10% of human genes are arranged in pairs divergently expressed from bidirectional promoters [7]. Most of them represent mRNA-lncRNA gene pairs with transcription start sites separated by an intergenic region not larger than 1 kb. The *FOXF1-FENDRR* gene pair constitute a different type of divergent gene arrangement in which the ~1.7 kb intergenic region contains two separate promoters (ENCODE candidate cis-regulatory elements cCREs: E1835413 and E1835410, respectively). Because of their close proximity, these promoters are thought to interact with each other, but the regulatory interactions involving head-to-head arranged genes are poorly understood. 

Intriguingly, in addition to causing ACDMPV, larger-sized CNV deletions involving *FOXF1* and its nearby genes *FENDRR*, *FOXC2,* and *FOXL1*, but not the pathogenic SNVs involving *FOXF1* or deletions of its distant lung-specific enhancer only, have been also associated with severe heart defects, including hypoplastic left heart syndrome (HLHS) and single umbilical artery (SUA) [2]. However, no HLHS-linked variants have been found in *FOXC2* and *FOXL1* ([8] and our unpublished data), whereas *Fendrr*^-/-^ mice developed defects in mesenchyme-derived tissues, including hypoplasia of the myocardium affecting ventricular walls [9] and/or the interventricular septum [9,10,11]. It has also been found that genetic variation in *FENDRR* is associated with an increased risk of developing hypertrophic cardiomyopathy [12]. Nevertheless, the genetic causes of HLHS or SUA remain unknown.

As is the case with other developmental genes, *FOXF1,* e.g., [13,14,15] and *FENDRR,* e.g., [16,17] are also involved in the pathogenesis of cancer; depending, e.g., on tissue context, they can function as tumor suppressors or oncogenes (FOXF1 as a transcription factor; *FENDRR* as a molecular sponge of miRNAs or scaffold/carrier for chromatin/mRNA modifying complexes). They both regulate cell proliferation, motility, and apoptosis. In addition, changes in the expression of *FENDRR* were causatively linked to lung and heart fibrosis, but the function of *FENDRR* in each of these organs is different, depends on the stage of development, and is still incompletely understood [17]. Deciphering the regulation of *FOXF1* and *FENDRR* is therefore important because of the possibility of using it to target these genes in future therapies for not only ACDMPV but also several more common disorders. 

Here, we describe a unique de novo CNV deletion that exclusively removed the paternal allele of *FOXF1* and its promoter, leaving the nearby *FENDRR* and its promoter intact, and causing only ACDMPV. We also functionally characterize the *FOXF1-FENDRR* bidirectional, two-promoter region using an in vitro reporter assay in the fetal pulmonary cell line. 

## 2. Results

### 2.1. Clinical Report

The male newborn (pt 219.3) was suspected of having ACDMPV based on the clinical course of his disease which was typical for this lethal developmental lung disorder (Appendix A).

### 2.2. CNV Deletion

The ACDMPV-causative CNV deletion was identified using whole genome sequencing (WGS). We mapped the CNV deletion breakpoints at chr16:86,509,520/1 and 86,514,272/3 (GRCh38/hg38) within non-repetitive sequences (Figure 1a, Appendix A). There was a 1 bp (C) microhomology at the deletion junction, suggesting that it might have arisen through a replication-based FoSTeS/MMBIR mechanism [18]. The deletion removed only *FOXF1* and its promoter (ENCODE cCRE E1835413), leaving the oppositely oriented nearby lncRNA gene *FENDRR*, its promoter (ENCODE cCRE E1835410), the enhancer-promoter interaction site [3], and the *FOXF1* & *FENDRR* TAD domain-forming two CTCF binding sites located 3–6 kb downstream of *FOXF1* [2], intact (Figure 1a).

Junction-specific PCR (Appendix A) in the parental DNA samples revealed that the deletion occurred de novo. There was no evidence of parental somatic mosaicism. Family trio Sanger sequencing of an informative SNP, rs11860845, amplified from the heterozygous CNV deletion region, showed that the CNV deletion arose in the proband on chr16q24.1 inherited from the father (Appendix A).

### 2.3. Functional Analysis of the Promoter Region Using Reporter Assay

To determine the transcriptional activity of the non-deleted portion of the *FOXF1* & *FENDRR* promoter region, harboring the putative *FENDRR* promoter (ENCODE cCRE E1835410), and to more systematically characterize the entire intergenic region, its fragments (Figure 1b) were PCR amplified from a normal human DNA, cloned into a promoter-less luciferase vector and analyzed for their promoter activity. We found that the non-deleted 672 bp part of the intergenic segment, containing the putative *FENDRR* promoter (construct A), exhibited transcriptional activity significantly above the reporter vector background (*p* = 0.01). This activity was similar to that of the deleted *FOXF1* promoter (ENCODE cCRE 1835413) (constructs E, F). Interestingly, in the absence of the *FOXF1* promoter, the region in between increased transcription measured from the *FENDRR* promoter 3-fold (*p* = 0.01; construct B). It exerted no effect on the *FOXF1* promoter both in the absence and presence of the *FENDRR* promoter (constructs E, F), and also exhibited its own transcriptional activity (construct D) (Figure 1b).

## 3. Discussion

We reported that paternal deletions at the *FOXF1* locus are 10 times less frequent than maternal ones, likely due to the proposed higher activity of the paternal allele of the *FOXF1* & *FENDRR* enhancer in extra-pulmonary tissues, resulting in more severe phenotypes and prenatal mortality [19]. Here, we hypothesize that the ACDMPV proband 219.3 did not manifest HLHS or SUA because the paternal deletion at chr16q24.1 did not involve *FENDRR* promoter and thus did not significantly affect *FENDRR* expression. Unfortunately, no biopsy or autopsy was performed and we could not test this hypothesis using lung tissue RNA. Instead, we did an in vitro reporter assay in fetal lung fibroblasts which showed that indeed the *FENDRR* promoter retained its activity.

Of note, two binding sites for BHLHE40 (ENCODE ChIP-seq), a transcription factor known to be involved in heart development and mesoderm commitment pathways [20], map to the undeleted portion of the *FOXF1-FENDRR* intergenic region (Figure 1). Based on the ENCODE ChIP-seq data, the *FENDRR* promoter also binds EP300, a major epigenetic regulator of chromatin involved in heart development [21].

To obtain a better insight into the functioning of the 1.7 kb bidirectional *FOXF1* & *FENDRR* promoter region, we systematically analyzed its activity using different PCR-generated overlapping fragments of this region (Figure 1b). We identified *FENDRR* expression-enhancing activity of the *FOXF1*-*FENDRR* inter-promoter region that was revealed only in the absence of the *FOXF1* promoter. This suggested that the inter-promoter region might interact with the *FOXF1* promoter (without affecting *FOXF1* promoter activity), but this interaction would suppress the ability of the inter-promoter region to positively regulate *FENDRR* (Figure 1c). Interestingly, ChIP-seq (ENCODE) revealed a significant binding of CTCF and the RAD21 cohesion subunit along *FOXF1* & *FENDRR* promoter region. These proteins may contribute to interactions between the inter-promoter segment and the *FOXF1* promoter, hindering for instance the interaction of the transcription factors, bound within the inter-promoter region, with the *FENDRR* promoter. Their interaction with the *FENDRR* promoter would become possible when the *FOXF1* promoter was deleted, or unable to interact with the inter-promoter region by other means. Recently, CTCF was also shown to suppress upstream antisense transcription independently of its function in maintaining chromatin architecture [22]. Alternatively, this inter-promoter region may harbor an additional *FENDRR* promoter that remains inactive in the presence of the *FOXF1* promoter.

Our data show the complexity of the *FOXF1*-*FENDRR* bidirectional two-promoter region whose inter-promoter segment, although being polarized towards activating *FENDRR*, is suppressed in cis by the *FOXF1* promoter. As transcription of the coding gene clearly predominates in most mRNA-lncRNA gene pairs, we hypothesize that a similar mechanism of lncRNA promoter suppression by mRNA gene promoter may be more prevalent with its net effect depending, e.g., on the distance between the promoters. The presented findings seem relevant, especially to other *FOX*-lncRNA divergent gene pairs. Our preliminary survey of the human genome/transcriptome (not shown) indicates that a significant fraction of all human *FOX* genes was arranged head-to-head with the neighboring lncRNA genes of unknown function. In some cases (e.g., *FOXC2*), the lncRNA gene partially overlapped with *FOX*. Lastly, understanding *FOXF1* and *FENDRR* regulation is important due to the essential role of these genes in the development and functioning of lungs, heart, and the digestive track. Their loss of function later in life may cause cancer or organ failure due to fibrosis. Therapeutic intervention in these diseases might include not only targeting *FOXF1* or *FENDRR* transcripts or inhibitors but *FOXF1-FENDRR* regulatory regions as well. Yet another interesting issue related to the deletion presented here is the genetics of some ACDMPV-associated disorders. With a caveat that the cell line used does not reflect the in vivo environment, we corroborate the previous notion on the possible causative link between lncRNA *FENDRR* and HLHS and SUA.

## 4. Materials and Methods

### 4.1. DNA Extraction and Sequencing

DNA was extracted from proband’s and his parents’ peripheral blood samples using Puregene Blood Core Kit (Qiagen, Hilden, Germany). WGS with 30x coverage was performed using a TruSeq Nano DNA HT Library Prep Kit (Illumina, San Diego, CA, USA) and the HiSeqX platform (Illumina, San Diego, CA, USA) at CloudHealth Genomics (Shanghai, China). The raw sequencing data were processed using bcl2fastq package (Illumina, San Diego, CA, USA) and Trimmomatic tool followed by read alignment and mapping to the human genome reference sequence with the BWA 0.7.12 tool. CNV deletion junction was amplified by long-range PCR using LA Taq DNApol (Takara Bio., Madison, WI, USA) and primer pair 5′-AGTGAGTGCGAACTTAAGCTCCTGTG-3′/5′-CCTCTATCTTTGG GAGTCTGGAAGGTAT-3′, directly Sanger sequenced, and aligned with the human genome GRCh37/hg19 using BLAT tool in the UCSC Genome Browser (https://genome.ucsc.edu accessed on 3 October 2023). 

### 4.2. Parental Origin of chr16 on Which the Deletion Was Found

Parental origin of the deletion-bearing chr16 was determined using informative SNPs amplified for Sanger sequencing from the proband’s DNA region of hemizygosity on chr16q24.1 and from corresponding region of parents’ chr16q24.1 with a primer pair 5′-GGGAGCGGGAAGTGACAAGA-3′/5′-CACAGATCGGTCTTTGTTTCAA-3′.

### 4.3. Luciferase Reporter Assay

PCR-amplified fragments of the *FOXF1* & *FENDRR* bidirectional promoter region (Figure 1b) were cloned into *Nhe*I-*Xho*I site within the multiple cloning sites of a luciferase (*luc2*) vector pGL4.10 (Promega, Madison, WI, USA). Primers used for amplifications and genomic coordinates of the amplicons are listed in Appendix A. PCR was performed using Phusion high-fidelity DNA pol (NEB, Ipswich, MA, USA), applying 30 cycles of incubation at 98 °C for 10 s, 58 °C for 30 s and 72 °C for 1.5 min. The amplified fragments were digested with *Kpn*I and *Nhe*I and cloned into *Kpn*I-*Nhe*I site of pGL4.10-*FOXF1*_promoter vector, upstream of *luc2*. For transfection, human fetal lung fibroblasts IMR-90 (ATCC, Manassas, VA, USA) were cultured at 37 °C in Eagle’s Minimal Essential Medium (ATCC), with 2 mM L-glutamine and 10% fetal bovine serum (ATCC), in the presence of 5% CO_2_ on 12-well plates. The cells were transfected in serum-free Opti-MEM (GIBCO, Waltham, MA, USA) using Lipofectamine 3000 (Invitrogen, Carlsbad, CA, USA) (4 µL/well) and 1 µg per well of the pGL4.10-*FOXF1*_promoter plasmid with or without (negative control) tested fragment, and 0.1 µg of pGL4.75 (Promega, Madison, WI, USA) constitutively expressing *Rluc* as a reference. RNA was isolated 48 h after transfection, using miRNeasy Mini Kit (Qiagen) and converted to cDNA using SuperScript III First-Strand Synthesis System (Invitrogen, Carlsbad, CA, USA).

The expression of *luc2* and *Rluc* was determined by measuring levels of cDNA of these two genes by qPCR. Custom-designed TaqMan primers and probes (*luc2*: assay AP7DRTC, amplicon coordinates in pGL4.10: 247–312, AY738222; and *Rluc*: assay AP47W76, amplicon coordinates in pGL4.75: 1532–1593, AY738231) were obtained from Applied Biosystems (Waltham, MA, USA). qPCRs were performed on CFX Real Time thermocycler (BioRad, Hercules, CA, USA) using TaqMan Universal PCR Master Mix (Applied Biosystems, Waltham, MA, USA). qPCR conditions included heating the reaction mixtures for 10 min at 95 °C, followed by 40 cycles of heating the mixtures for 15 s at 95 °C and 1 min at 60 °C. For relative quantification of the cDNAs/transcripts, the comparative CT method was used. *luc2* cDNA levels were normalized to those of *Rluc*.

## Figures and Tables

**Figure 1 ncrna-09-00061-f001:**
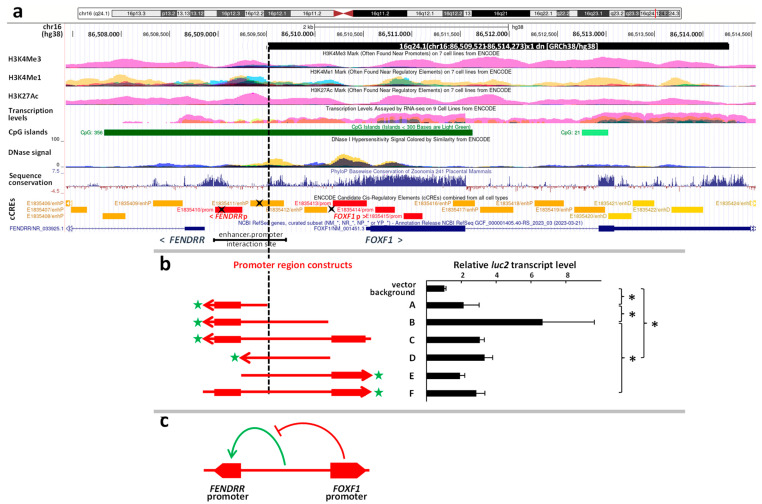
Schematic representation of the structure and function of the *FOXF1* & *FENDRR* bidirectional promoter region. (**a**) The novel ACDMPV-causative CNV deletion of *FOXF1* at chr16q24.1 (black bar), ENCODE promoter cCREs (red bars), C4-determined [3] distant enhancer-promoter interaction site, ChIP-seq-determined (ENCODE) BHLHE40 transcription factor binding sites (black crosses) are shown. Histone H3 modification signal and DNase I hypersensitivity signal in lung fibroblast are depicted in pink and yellow, respectively. (**b**) Functional dissection of the *FOXF1* & *FENDRR* promoter region. The left panel shows the promoter regions tested using the luciferase reporter assay, aligned with the genomic region in (**a**). The orientation of each fragment versus the reporter gene (green stars) in the pGL4.10 vector is indicated by an arrow. Red bars correspond to the promoter cCREs as in (**a**). The right panel shows relative transcriptional activity of the promoter fragments from the left panel (error-bars represent the standard deviation; * *p* = 0.01). The assay indicates that the described CNV deletion might have not affected significantly the activity of the *FENDRR* promoter. It also unmasked, silenced in the presence of the *FOXF1* promoter, polarization towards activating *FENDRR* of the inter-promoter region. (**c**) Model of the interaction between the *FOXF1* and *FENDRR* promoters.

## Data Availability

The data and materials are available from the corresponding authors upon reasonable request.

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
