# Peer review of "A Small De Novo CNV Deletion of the Paternal Copy of *FOXF1*, Leaving lncRNA *FENDRR* Intact, Provides Insight into Their Bidirectional Promoter Region"

_ncrna, 2023, doi:10.3390/ncrna9050061_

Round 1

Reviewer 1 Report

Dear authors,

After consideration, I have some recommendations and concerns that I believe will improve the quality and impact of your manuscript.

While the manuscript provides a good overview of the study's context and objectives, the introduction could be enhanced by providing more background information. Specifically, I suggest incorporating additional details on the significance of bi-directional promoter regions in gene regulation and possibly discussing relevant literature demonstrating the importance of FOXF1 and lncRNA FENDRR in biological processes. A more comprehensive introduction will provide readers with a clearer understanding of the broader implications of your research.

Figure 1a has low resolution, which can affect the clarity and interpretation of the data presented. You should provide a higher-resolution version since it is essential for readers to appreciate the details of your findings.

To strengthen the generalizability of your findings, I suggest you include examples or references to other genes or genomic regions where similar bi-directional promoter regulation mechanisms have been observed. This could help establish the broader relevance of your study and highlight the uniqueness of your findings in the context of existing research.

Given the emerging role of lncRNA FENDRR in gene regulation and its potential involvement in disease pathways, particularly in cancer development, I suggest that you briefly discuss the possible implications of your findings in this regard, exploring the connections between FENDRR and cancer development, supported by relevant literature or hypotheses, enhancing the significance of your research.

I believe that addressing these points will significantly improve your manuscript's overall quality and impact.

          Sincerely,

Author Response

Reviewer 1

After consideration, I have some recommendations and concerns that I believe will improve the quality and impact of your manuscript.

While the manuscript provides a good overview of the study's context and objectives, the introduction could be enhanced by providing more background information. Specifically, I suggest incorporating additional details on the significance of bi-directional promoter regions in gene regulation and possibly discussing relevant literature demonstrating the importance of FOXF1 and lncRNA FENDRR in biological processes. A more comprehensive introduction will provide readers with a clearer understanding of the broader implications of your research.

  • We thank the Reviewer for the constructive comments. In the revised manuscript, we have added two paragraphs to the Introduction, one dealing with bidirectional promoters:

Eukaryotic promoters are intrinsically bidirectional featuring two transcription start sites; however, transcription in one direction usually prevails [6]. Nevertheless, as much as 10% of human genes are arranged in pairs divergently expressed from bidirectional promoters [7]. Most of them represent mRNA-lncRNA gene pairs with transcription start sites separated by an intergenic region not larger than 1 kb. The FOXF1-FENDRR gene pair constitute a different type of divergent gene arrangement in which ~ 1.7 kb intergenic region contains two separate promoters (ENCODE candidate cis-regulatory elements, cCREs: E1835413 and E1835410, respectively). Because of their close proximity, these promoters are thought to interact with each other, but the regulatory interactions involving head-to-head arranged genes are in general poorly understood.

and the other emphasizing the importance of FOXF1 and FENDRR:

As is the case with other developmental genes, FOXF1 [e.g., 13-15] and FENDRR [e.g., 16,17] are also involved in the pathogenesis of cancer, depending e.g. on tissue context, as tumor suppressors or oncogenes (FOXF1 as a transcription factor; FENDRR as a molecular sponge of miRNAs or scaffold/carrier for chromatin/mRNA modifying complexes). They both regulate cell proliferation, motility, and apoptosis. In addition, changes in expression of FENDRR were causatively linked to lung and heart fibrosis, but the function of FENDRR in each of these organs is different and still incompletely understood [17]. Deciphering the regulation of FOXF1 and FENDRR is therefore important because of the potential of both these genes in the development of therapies for several common disorders.

Figure 1a has low resolution, which can affect the clarity and interpretation of the data presented. You should provide a higher-resolution version since it is essential for readers to appreciate the details of your findings.

  • We have improved the quality of the Fig. 1 (on the revised image saved as PDF file, all lettering is easily visible at least at higher magnification; only the image embedded in the text has lower resolution).

To strengthen the generalizability of your findings, I suggest you include examples or references to other genes or genomic regions where similar bi-directional promoter regulation mechanisms have been observed. This could help establish the broader relevance of your study and highlight the uniqueness of your findings in the context of existing research.

  • We included information on other divergent gene pairs in the Introduction:

“…, as much as 10% of human genes are arranged in pairs divergently expressed from a bidirectional promoter [19].”

and in the Discussion:

“The presented findings seem relevant especially to other FOX-lncRNA divergent gene pairs. Our preliminary survey of human genome/transcriptome (not shown) indicate that a significant fraction of allhuman FOX genes was arranged head-to-head with the neighboring lncRNA genes of unknown function. In some cases (e.g., FOXC2) the lncRNA gene partially overlapped with FOX.”

We have also added two references.

Given the emerging role of lncRNA FENDRR in gene regulation and its potential involvement in disease pathways, particularly in cancer development, I suggest that you briefly discuss the possible implications of your findings in this regard, exploring the connections between FENDRR and cancer development, supported by relevant literature or hypotheses, enhancing the significance of your research.

  • We have added a new paragraph on the role of FOXF1 and FENDRR in cancer and fibrosis in the revised Introduction section:

As is the case with other developmental genes, FOXF1 [e.g., 13-15] and FENDRR [e.g., 16,17] are also involved in the pathogenesis of cancer, depending e.g. on tissue context, as tumor suppressors or oncogenes (FOXF1 as a transcription factor; FENDRR as a molecular sponge of miRNAs or scaffold/carrier for chromatin/mRNA modifying complexes). They both regulate cell proliferation, motility and apoptosis. In addition, changes in expression of FENDRR were causatively linked to lung and heart fibrosis, but the function of FENDRR in each of these organs is different and still incompletely understood [17]. Deciphering the regulation of FOXF1 and FENDRR is therefore important because of the potential of both these genes in the development of therapies for several common disorders.

We have also added five references.

I believe that addressing these points will significantly improve your manuscript's overall quality and impact.

Reviewer 2 Report

In the manuscript Szafranski and Stankiewicz describe an interesting case report of a deletion of FOXF1 and its promoter in an infant with ACDMPV. They report that the neighboring non-coding transcript FENDRR is left intact and expressed. Reporter assay showed transcriptional activity of the retained FENDRR promoter. Altogether, it is an interesting report, highlighting the potential mechanism of interaction between a protein-coding and non-coding gene promoters, which would however require further validation. I recommend to improve the resolution of Figure 1 as it is illegible after zooming in.

Author Response

Reviewer 2

In the manuscript Szafranski and Stankiewicz describe an interesting case report of a deletion of FOXF1 and its promoter in an infant with ACDMPV.

  • We thank the Reviewer for the interest in our work.

They report that the neighboring non-coding transcript FENDRR is left intact and expressed. Reporter assay showed transcriptional activity of the retained FENDRR promoter. Altogether, it is an interesting report, highlighting the potential mechanism of interaction between a protein-coding and non-coding gene promoters, which would however require further validation. I recommend to improve the resolution of Figure 1 as it is illegible after zooming in.

  • We have improved the quality of the revised Fig. 1.